# Tagger: Deep Unsupervised Perceptual Grouping

**Klaus Greff**[*], **Antti Rasmus, Mathias Berglund, Tele Hotloo Hao,**
**Jürgen Schmidhuber**[*], **Harri Valpola**
The Curious AI Company {antti,mathias,hotloo,harri}@cai.fi
[*]IDSIA {klaus,juergen}@idsia.ch

## Abstract

We present a framework for efficient perceptual inference that explicitly reasons about the segmentation of its inputs and features. Rather than being trained for any specific segmentation, our framework learns the grouping process in an unsupervised manner or alongside any supervised task. We enable a neural network to group the representations of different objects in an iterative manner through a differentiable mechanism. We achieve very fast convergence by allowing the system to amortize the joint iterative inference of the groupings and their representations. In contrast to many other recently proposed methods for addressing multi-object scenes, our system does not assume the inputs to be images and can therefore directly handle other modalities. We evaluate our method on multi-digit classification of very cluttered images that require texture segmentation. Remarkably our method achieves improved classification performance over convolutional networks despite being fully connected, by making use of the grouping mechanism. Furthermore, we observe that our system greatly improves upon the semi-supervised result of a baseline Ladder network on our dataset. These results are evidence that grouping is a powerful tool that can help to improve sample efficiency.

## 1 Introduction

Humans naturally perceive the world as being structured into different objects, their properties and relation to each other. This phenomenon which we refer to as perceptual grouping is also known as amodal perception in psychology. It occurs effortlessly and includes a segmentation of the visual input, such as that shown in in Figure 1. This grouping also applies analogously to other modalities, for example in solving the cocktail party problem (audio) or when separating the sensation of a grasped object from the sensation of fingers touching each other (tactile). Even more abstract features such as object class, color, position, and velocity are naturally grouped together with the inputs to form coherent objects. This rich structure is crucial for many real-world tasks such as manipulating objects or driving a car, where awareness of different objects and their features is required.

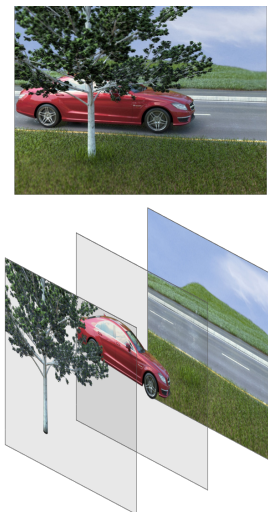

Figure 1: An example of perceptual grouping for vision.

In this paper, we introduce a framework for learning efficient iterative inference of such perceptual grouping which we call *iTerative Amortized Grouping* (TAG). This framework entails a mechanism for iteratively splitting the inputs and internal representations into several different groups. We make no assumptions about the structure of this segmentation and rather train the model end-to-end to discover which are the relevant features and how to perform the splitting.

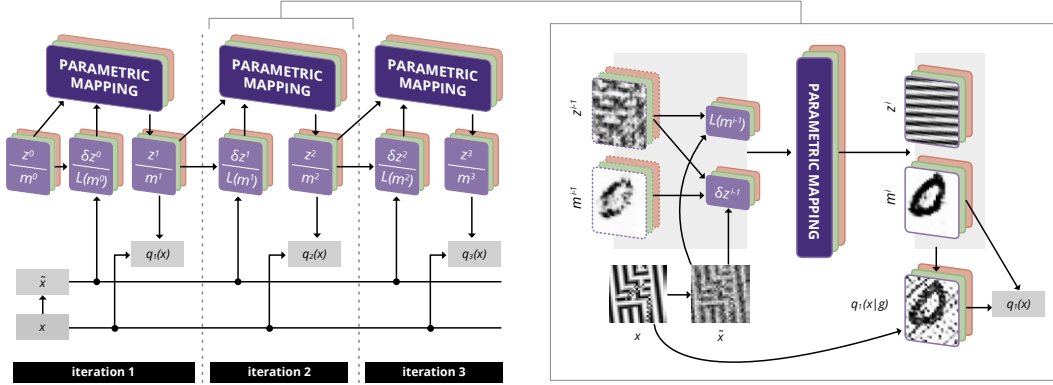

Figure 2: Left: Three iterations of the TAG system which learns by denoising its input using several groups (shown in color). Right: Detailed view of a single iteration on the TextureMNIST1 dataset. Please refer to the supplementary material for further details.

By using an auxiliary denoising task we train the system to directly amortize the posterior inference of the object features and their grouping. Because our framework does not make any assumptions about the structure of the data, it is completely domain agnostic and applicable to any type of data. The TAG framework works completely unsupervised, but can also be combined with supervised learning for classification or segmentation.

## 2  Iterative Amortized Grouping (TAG)[1]

**Grouping.**  Our goal is to enable neural networks to split inputs and internal representations into coherent *groups*. We define a group to be a collection of inputs and internal representations that are processed together, but (largely) independent of each other. By processing each group separately the network can make use of invariant distributed features without the risk of interference and ambiguities, which might arise when processing everything in one clump. We make no assumptions about the correspondence between objects and groups. If the network can process several objects in one group without unwanted interference, then the network is free to do so. The "correct" grouping is often dynamic, ambiguous and task dependent. So rather than training it as a separate task, we allow the network to split the processing of the inputs, and let it learn how to best use this ability for any given problem. To make the task of instance segmentation easy, we keep the groups symmetric in the sense that each group is processed by the same underlying model.

**Amortized Iterative Inference.**  We want our model to reason not only about the group assignments but also about the representation of each group. This amounts to inference over two sets of variables: the latent group assignments and the individual group representations; A formulation very similar to mixture models for which exact inference is typically intractable. For these models it is a common approach to approximate the inference in an iterative manner by alternating between (re-)estimation of these two sets (e.g., EM-like methods [4]). The intuition is that given the grouping, inferring the object features becomes easy, and vice versa. We employ a similar strategy by allowing our network to iteratively refine its estimates of the group assignments as well as the object representations.

Rather than deriving and then running an inference algorithm, we train a parametric mapping to arrive at the end result of inference as efficiently as possible [9]. This is known as *amortized* inference [31], and it is used, for instance, in variational autoencoders where the encoder learns to amortize the posterior inference required by the generative model represented by the decoder. Here we instead apply the framework of denoising autoencoders [6, 15, 34] which are trained to reconstruct original inputs $x$ from corrupted versions $\tilde{x}$. This encourages the network to implement useful amortized posterior inference without ever having to specify or even know the underlying generative model whose inference is implicitly learned.

**Data**: $\boldsymbol{x}, K, T, \sigma, v, W_h, W_u, \Theta$
**Result**: $\boldsymbol{z}^T, \boldsymbol{m}^T, C$
**begin** Initialization:
 $\quad \tilde{\boldsymbol{x}} \leftarrow \boldsymbol{x} + \mathcal{N}(\boldsymbol{0}, \sigma^2 \boldsymbol{I})$;
 $\quad \boldsymbol{m}^0 \leftarrow \text{softmax}(\mathcal{N}(\boldsymbol{0}, \boldsymbol{I}))$;
 $\quad \boldsymbol{z}^0 \leftarrow E[\boldsymbol{x}]$;
**end**
**for** $i = 0 \ldots T-1$ **do**
 **for** $k = 1 \ldots K$ **do**
 $\quad \tilde{\boldsymbol{z}}_k \leftarrow \mathcal{N}(\tilde{\boldsymbol{x}}; \boldsymbol{z}_k^i, (v + \sigma^2)\boldsymbol{I})$;
 $\quad \boldsymbol{\delta z_k^i} \leftarrow (\tilde{\boldsymbol{x}} - \boldsymbol{z}_k^i)\boldsymbol{m}_k^i \tilde{\boldsymbol{z}}_k$;
 $\quad L(\boldsymbol{m}_k^i) \leftarrow \frac{\tilde{\boldsymbol{z}}_k}{\sum_h \tilde{\boldsymbol{z}}_h}$ ;
 $\quad \boldsymbol{h}_k^i \leftarrow f(W_h \left[\boldsymbol{z}_k^i, \boldsymbol{m}_k^i, \boldsymbol{\delta z}_k^i, L(\boldsymbol{m}_k^i)\right])$;
 $\quad [\boldsymbol{z}_k^{i+1}, \boldsymbol{m}_k^{i+1}] \leftarrow W_u \text{Ladder}(\boldsymbol{h}_k^i, \Theta)$;
 **end**
 $\quad \boldsymbol{m}^{i+1} \leftarrow \text{softmax}(\boldsymbol{m}^{i+1})$;
 $\quad q_{i+1}(\boldsymbol{x}) \leftarrow \sum_{k=1}^K \mathcal{N}(\boldsymbol{x}; \boldsymbol{z}_k^{i+1}, v\boldsymbol{I})\boldsymbol{m}^{i+1}$;
**end**
$C \leftarrow -\sum_{i=1}^T \log q_i(\boldsymbol{x})$;

**Algorithm 1:** Pseudocode for running Tagger on a single real-valued example $\boldsymbol{x}$. For details and a binary-input version please refer to supplementary material.

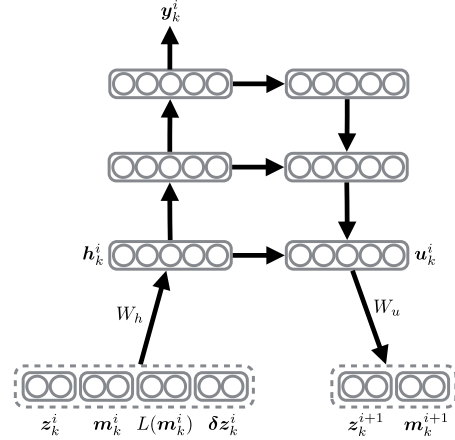

Figure 3: An example of how Tagger would use a 3-layer-deep Ladder Network as its parametric mapping to perform its iteration $i+1$. Note the optional class prediction output $\boldsymbol{y}_g^i$ for classification tasks. See supplementary material for details.

**Putting it together.**  By using the negative log likelihood $C(\boldsymbol{x}) = -\sum_i \log q_i(\boldsymbol{x})$ as a cost function, we train our system to compute an approximation $q_i(\boldsymbol{x})$ of the true denoising posterior $p(\boldsymbol{x}|\tilde{\boldsymbol{x}})$ at each iteration $i$. An overview of the whole system is given in Figure 2. For each input element $x_j$ we introduce $K$ latent binary variables $g_{k,j}$ that take a value of 1 if this element is generated by group $k$. This way inference is split into $K$ groups, and we can write the approximate posterior in vector notation as follows:

$$q_i(\boldsymbol{x}) = \sum_k q_i(\boldsymbol{x}|\boldsymbol{g}_k)q_i(\boldsymbol{g}_k) = \sum_k \mathcal{N}(\boldsymbol{x}; \boldsymbol{z}_k^i, v\boldsymbol{I})\boldsymbol{m}_k^i \ , \tag{1}$$

where we model the group reconstruction $q_i(\boldsymbol{x}|\boldsymbol{g_k})$ as a Gaussian with mean $\boldsymbol{z}_k^i$ and variance $v$, and the group assignment posterior $q_i(\boldsymbol{g_k})$ as a categorical distribution $\boldsymbol{m}_k$.

The trainable part of the TAG framework is given by a parametric mapping that operates independently on each group $k$ and is used to compute both $\boldsymbol{z}_k^i$ and $\boldsymbol{m}_k^i$ (which is afterwards normalized using an elementwise softmax over the groups). This parametric mapping is usually implemented by a neural network and the whole system is trained end-to-end using standard backpropagation through time.

The input to the network for the next iteration consists of the vectors $\boldsymbol{z}_k^i$ and $\boldsymbol{m}_k^i$ along with two additional quantities: The remaining modelling error $\boldsymbol{\delta z}_k^i$ and the group assignment likelihood ratio $L(\boldsymbol{m}_k^i)$ which carry information about how the estimates can be improved:

$$\boldsymbol{\delta z}_k^i \propto \frac{\partial C(\tilde{\boldsymbol{x}})}{\partial \boldsymbol{z}_k^i} \qquad \text{and} \qquad L(\boldsymbol{m}_k^i) \propto \frac{q_i(\tilde{\boldsymbol{x}}|\boldsymbol{g}_k)}{\sum_h q_i(\tilde{\boldsymbol{x}}|\boldsymbol{g}_h)}$$

Note that they are derived from the corrupted input $\tilde{\boldsymbol{x}}$, to make sure we don't leak information about the clean input $\boldsymbol{x}$ into the system.

**Tagger.**  For this paper we chose the Ladder network [19] as the parametric mapping because its structure reflects the computations required for posterior inference in hierarchical latent variable models. This means that the network should be well equipped to handle the hierarchical structure one might expect to find in many domains. We call this Ladder network wrapped in the TAG framework *Tagger*. This is illustrated in Figure 3 and the corresponding pseudocode can be found in Algorithm 1.

# 3 Experiments and results

We explore the properties and evaluate the performance of Tagger both in fully unsupervised settings and in semi-supervised tasks in two datasets[2]. Although both datasets consist of images and grouping is intuitively similar to image segmentation, there is no prior in the Tagger model for images: our results (unlike the ConvNet baseline) generalize even if we permute all the pixels .

**Shapes.** We use the simple Shapes dataset [21] to examine the basic properties of our system. It consists of 60,000 (train) + 10,000 (test) binary images of size 20x20. Each image contains three randomly chosen shapes ($\triangle$, $\triangledown$, or $\square$) composed together at random positions with possible overlap.

**Textured MNIST.** We generated a two-object supervised dataset (TextureMNIST2) by sequentially stacking two textured 28x28 MNIST-digits, shifted two pixels left and up, and right and down, respectively, on top of a background texture. The textures for the digits and background are different randomly shifted samples from a bank of 20 sinusoidal textures with different frequencies and orientations. Some examples from this dataset are presented in the column of Figure 4b. We use a 50k training set, 10k validation set, and 10k test set to report the results. We also use a textured single-digit version (TextureMNIST1) without a shift to isolate the impact of texturing from multiple objects.

## 3.1 Training and evaluation

We train Tagger in an unsupervised manner by only showing the network the raw input example $x$, not ground truth masks or any class labels, using 4 groups and 3 iterations. We average the cost over iterations and use ADAM [14] for optimization. On the Shapes dataset we trained for 100 epochs with a bit-flip probability of 0.2, and on the TextureMNIST dataset for 200 epochs with a corruption-noise standard deviation of 0.2. The models reported in this paper took approximately 3 and 11 hours in wall clock time on a single Nvidia Titan X GPU for Shapes and TextureMNIST2 datasets respectively.

We evaluate the trained models using two metrics: First, the denoising cost on the validation set, and second we evaluate the segmentation into objects using the *adjusted mutual information (AMI) score* [35] and ignore the background and overlap regions in the Shapes dataset (consistent with Greff et al. [8]). Evaluations of the AMI score and classification results in semi-supervised tasks were performed using uncorrupted input. The system has no restrictions regarding the number of groups and iterations used for training and evaluation. The results improved in terms of both denoising cost and AMI score when iterating further, so we used 5 iterations for testing. Even if the system was trained with 4 groups and 3 shapes per training example, we could test the evaluation with, for example, 2 groups and 3 shapes, or 4 groups and 4 shapes.

## 3.2 Unsupervised Perceptual Grouping

Table 1 shows the median performance of Tagger on the Shapes dataset over 20 seeds. Tagger is able to achieve very fast convergences, as shown in Table 1a. Through iterations, the network improves its denoising performances by grouping different objects into different groups. Comparing to Greff et al. [8], Tagger performs significantly better in terms of AMI score (see Table 1b). We found that for this dataset using LayerNorm [1] instead of BatchNorm [13] greatly improves the results as seen in Table 1.

Figure 4a and Figure 4b qualitatively show the learned unsupervised groupings for the Shapes and textured MNIST datasets. Tagger uses its TAG mechanism slightly differently for the two datasets. For Shapes, $z_g$ represents filled-in objects and masks $m_g$ show which part of the object is actually visible. For textured MNIST, $z_g$ represents the textures while masks $m_g$ capture texture segments. In the case of the same digit or two identical shapes, Tagger can segment them into separate groups, and hence, performs *instance segmentation*. We used 4 groups for training even though there are only 3 objects in the Shapes dataset and 3 segments in the TexturedMNIST2 dataset. The excess group is left empty by the trained system but its presence seems to speed up the learning process.

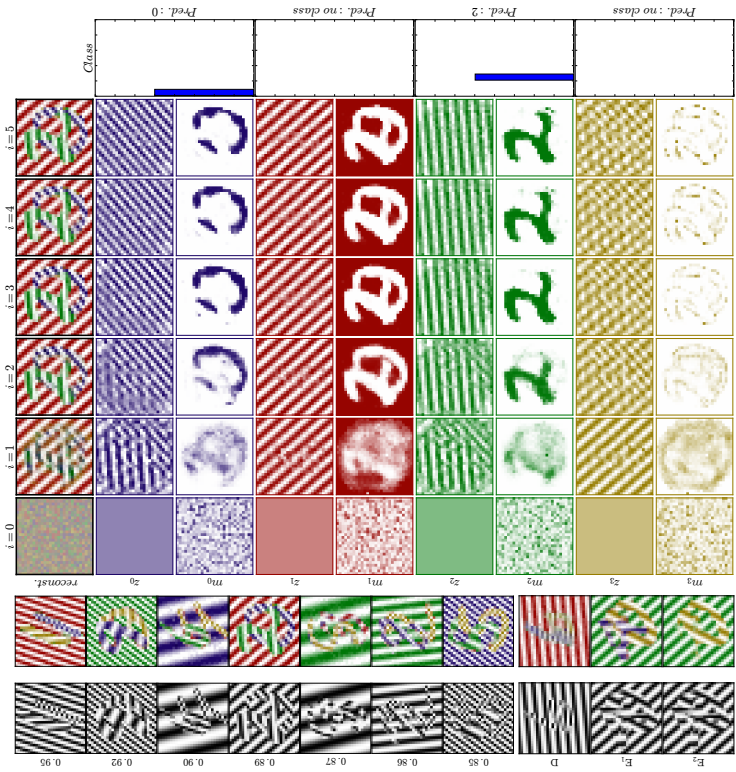

(b) Results for the TextureMNIST2 dataset. Left column: 7 examples from the test set along with their resulting groupings in descending AMI score order and 3 hand-picked examples (D, E1, E2). D: An example from the TextureMNIST1 dataset. E1-2: A hand-picked example from TextureMNIST2. E1 demonstrates typical inference, and E2 demonstrates how the system is able to estimate the input when a certain group (topmost digit 4) is removed. Right column: Illustration of the inference process over iterations for four color-coded groups; $m_k$ and $z_k$.

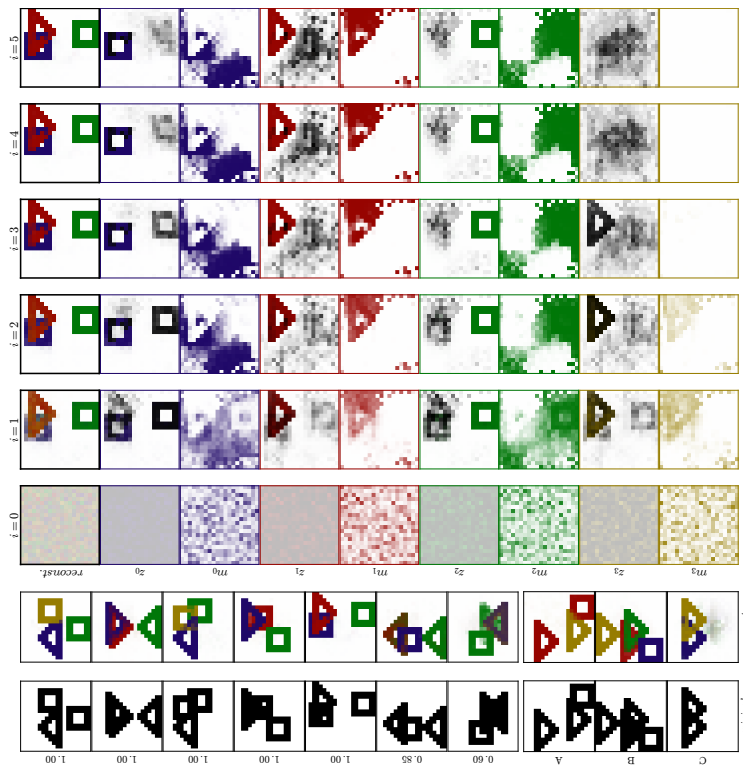

(a) Results for Shapes dataset. Left column: 7 examples from the test set along with their resulting groupings in descending AMI score order and 3 hand-picked examples (A, B, and C) to demonstrate generalization. A: Testing 2-group model on 3 object data. B: Testing a 4-group model trained with 3-object data on 4 objects. C: Testing 4-group model trained with 3-object data on 2 objects. Right column: Illustration of the inference process over iterations for four color-coded groups; $m_k$ and $z_k$.

|  | Iter 1 | Iter 2 | Iter 3 | Iter 4 | Iter 5 |
|---|---|---|---|---|---|
| Denoising cost | 0.094 | 0.068 | 0.063 | 0.063 | 0.063 |
| AMI | 0.58 | 0.73 | 0.77 | 0.79 | 0.79 |
| Denoising cost* | 0.100 | 0.069 | 0.057 | **0.054** | **0.054** |
| AMI* | 0.70 | 0.90 | 0.95 | 0.96 | **0.97** |

(a) Convergence of Tagger over iterative inference

|  | AMI |
|---|---|
| RC [8] | $0.61 \pm 0.005$ |
| Tagger | $0.79 \pm 0.034$ |
| Tagger* | **$0.97 \pm 0.009$** |

(b) Method comparison

Table 1: Table (a) shows how quickly the algorithm evaluation converges over inference iterations with the Shapes dataset. Table (b) compares segmentation quality to previous work on the Shapes dataset. The AMI score is defined in the range from 0 (guessing) to 1 (perfect match). The results with a star (*) are using LayerNorm [1] instead of BatchNorm.

The hand-picked examples A-C in Figure 4a illustrate the robustness of the system when the number of objects changes in the evaluation dataset or when evaluation is performed using fewer groups. Example $E$ is particularly interesting; $E_2$ demonstrates how we can remove the topmost digit from the normal evaluated scene $E_1$ and let the system fill in digit below and the background. We do this by setting the corresponding group assignment probabilities $m_g$ to a large negative number just before the final softmax over groups in the last iteration.

To solve the textured two-digit MNIST task, the system has to combine texture cues with high-level shape information. The system first infers the background texture and mask which are finalized on the first iteration. Then the second iteration typically fixes the texture used for topmost digit, while subsequent iterations clarify the occluded digit and its texture. This demonstrates the need for iterative inference of the grouping.

## 3.3 Classification

To investigate the role of grouping for the task of classification, we evaluate Tagger against four baseline models on the textured MNIST task. As our first baseline we use a fully connected network (*FC*) with ReLU activations and BatchNorm [13] after each layer. Our second baseline is a ConvNet (*Conv*) based on Model C from [30], which has close to state-of-the-art results on CIFAR-10. We removed dropout, added BatchNorm after each layer and replaced the final pooling by a fully connected layer to improve its performance for the task. Furthermore, we compare with a fully connected Ladder [19] (FC Ladder) network.

All models use a softmax output and are trained with 50,000 samples to minimize the categorical cross entropy error. In case there are two different digits in the image (most examples in the TextureMNIST2 dataset), the target is $p = 0.5$ for both classes. We evaluate the models based on classification errors, which we compute based on the two highest predicted classes (top 2) for the two-digit case.

For Tagger, we first train the system in an unsupervised phase for 150 epochs and then add two fresh randomly initialized layers on top and continue training the entire system end to end using the sum of unsupervised and supervised cost terms for 50 epochs. Furthermore, the topmost layer has a per-group softmax activation that includes an added 'no class' neuron for groups that do not contain any digit. The final classification is then performed by summing the softmax output over all groups for the true 10 classes and renormalizing it.

As shown in Table 2, Tagger performs significantly better than all the fully connected baseline models on both variants, but the improvement is more pronounced for the two-digit case. This result is expected because for cases with multi-object overlap, grouping becomes more important. Moreover, it confirms the hypothesis that grouping can help classification and is particularly beneficial for complex inputs. Remarkably, Tagger is on par with the convolutional baseline for the TexturedMNIST1 dataset and even outperforms it in the two-digit case, despite being fully connected itself. We hypothesize that one reason for this result is that grouping allows for the construction of efficient invariant features already in the low layers without losing information about the assignment of features to objects. Convolutional networks solve this problem to some degree by grouping features locally through the use of receptive fields, but that strategy is expensive and can break down in cases of heavy overlap.

| Dataset | Method | Error 50k | Error 1k | Model details |
|---|---|---|---|---|
| TextureMNIST1 | FC MLP | $31.1 \pm 2.2$ | $89.0 \pm 0.2$ | 2000-2000-2000 / 1000-1000 |
| *chance level: 90%* | FC Ladder | $7.2 \pm 0.1$ | $30.5 \pm 0.5$ | 3000-2000-1000-500-250 |
| | FC Tagger (ours) | $\mathbf{4.0 \pm 0.3}$ | $\mathbf{10.5 \pm 0.9}$ | 3000-2000-1000-500-250 |
| | *ConvNet* | $\mathbf{3.9 \pm 0.3}$ | $52.4 \pm 5.3$ | based on Model C [30] |
| TextureMNIST2 | FC MLP | $55.2 \pm 1.0$ | $79.4 \pm 0.3$ | 2000-2000-2000 / 1000-1000 |
| *chance level: 80%* | FC Ladder | $41.1 \pm 0.2$ | $68.5 \pm 0.2$ | 3000-2000-1000-500-250 |
| | FC Tagger (ours) | $\mathbf{7.9 \pm 0.3}$ | $\mathbf{24.9 \pm 1.8}$ | 3000-2000-1000-500-250 |
| | *ConvNet* | $12.6 \pm 0.4$ | $79.1 \pm 0.8$ | based on Model C [30] |

Table 2: Test-set classification errors in % for both textured MNIST datasets. We report mean and sample standard deviation over 5 runs. FC = Fully Connected, MLP = Multi Layer Perceptron.

### 3.4 Semi-Supervised Learning

The TAG framework does not rely on labels and is therefore directly usable in a semi-supervised context. For semi-supervised learning, the Ladder [19] is arguably one of the strongest baselines with SOTA results on 1,000 MNIST and 60,000 permutation invariant MNIST classification. We follow the common practice of using 1,000 labeled samples and 49,000 unlabeled samples for training Tagger and the Ladder baselines. For completeness, we also report results of the convolutional (*ConvNet*) and fully-connected (*FC*) baselines trained fully supervised on only 1,000 samples.

From Table 2, it is obvious that all the fully supervised methods fail on this task with 1,000 labels. The best baseline result is achieved by the *FC Ladder*, which reaches 30.5 % error for one digit but 68.5 % for TextureMNIST2. For both datasets, Tagger achieves by far the lowest error rates: 10.5 % and 24.9 %, respectively. Again, this difference is amplified for the two-digit case, where Tagger with 1,000 labels even outperforms the Ladder baseline with all 50k labels. This result matches our intuition that grouping can often segment even objects of an unknown class and thus help select the relevant features for learning. This is particularly important in semi-supervised learning where the inability to self-classify unlabeled samples can mean that the network fails to learn from them at all.

To put these results in context, we performed informal tests with five human subjects. The subjects improved significantly over training for a few days but there were also significant individual differences. The task turned out to be quite difficult and strenuous, with the best performing subjects scoring around 10 % error for TextureMNIST1 and 30 % error for TextureMNIST2.

## 4 Related work

Attention models have recently become very popular, and similar to perceptual grouping they help in dealing with complex structured inputs. These approaches are not, however, mutually exclusive and can benefit from each other. Overt attention models [28, 5] control a window (fovea) to focus on relevant parts of the inputs. Two of their limitations are that they are mostly tailored to the visual domain and are usually only suited to objects that are roughly the same shape as the window. But their ability to limit the field of view can help to reduce the complexity of the target problem and thus also help segmentation. Soft attention mechanisms [26, 3, 40] on the other hand use some form of top-down feedback to suppress inputs that are irrelevant for a given task. These mechanisms have recently gained popularity, first in machine translation [2] and then for many other problems such as image caption generation [39]. Because they re-weigh all the inputs based on their relevance, they could benefit from a perceptual grouping process that can refine the precise boundaries of attention.

Our work is primarily built upon a line of research based on the concept that the brain uses synchronization of neuronal firing to bind object representations together. This view was introduced by [37] and has inspired many early works on oscillations in neural networks (see the survey [36] for a summary). Simulating the oscillations explicitly is costly and does not mesh well with modern neural network architectures (but see [17]). Rather, complex values have been used to model oscillating activations using the phase as soft tags for synchronization [18, 20]. In our model, we further abstract them by using discretized synchronization slots (our groups). It is most similar to the models of Wersing et al. [38], Hyvärinen & Perkiö [12] and Greff et al. [8]. However, our work is the first to combine this with denoising autoencoders in an end-to-end trainable fashion.

Another closely related line of research [23, 22] has focused on multi-causal modeling of the inputs. Many of the works in that area [16, 32, 29, 11] build upon Restricted Boltzmann Machines. Each input is modeled as a mixture model with a separate latent variable for each object. Because exact inference is intractable, these models approximate the posterior with some form of expectation maximization [4] or sampling procedure. Our assumptions are very similar to these approaches, but we allow the model to learn the amortized inference directly (more in line with Goodfellow et al. [7]).

Since recurrent neural networks (RNNs) are general purpose computers, they can in principle implement arbitrary computable types of temporary variable binding [25, 26], unsupervised segmentation [24], and internal [26] and external attention [28]. For example, an RNN with fast weights [26] can rapidly associate or bind the patterns to which the RNN currently attends. Similar approaches even allow for metalearning [27], that is, learning a learning algorithm. Hochreiter et al. [10], for example, learned fast online learning algorithms for the class of all quadratic functions of two variables. Unsupervised segmentation could therefore in principle be learned by any RNN as a by-product of data compression or any other given task. That does not, however, imply that every RNN will, through learning, easily discover and implement this tool. From that perspective, TAG can be seen as a way of helping an RNN to quickly learn and efficiently implement a grouping mechanism.

## 5   Conclusion

In this paper, we have argued that the ability to group input elements and internal representations is a powerful tool that can improve a system's ability to handle complex multi-object inputs. We have introduced the TAG framework, which enables a network to directly learn the grouping and the corresponding amortized iterative inference in a unsupervised manner. The resulting iterative inference is very efficient and converges within five iterations. We have demonstrated the benefits of this mechanism for a heavily cluttered classification task, in which our fully connected Tagger even significantly outperformed a state-of-the-art convolutional network. More impressively, we have shown that our mechanism can greatly improve semi-supervised learning, exceeding conventional Ladder networks by a large margin. Our method makes minimal assumptions about the data and can be applied to any modality. With TAG, we have barely scratched the surface of a comprehensive integrated grouping mechanism, but we already see significant advantages. We believe grouping to be crucial to human perception and are convinced that it will help to scale neural networks to even more complex tasks in the future.

### Acknowledgments

The authors wish to acknowledge useful discussions with Theofanis Karaletsos, Jaakko Särelä, Tapani Raiko, and Søren Kaae Sønderby. And further acknowledge Rinu Boney, Timo Haanpää and the rest of the Curious AI Company team for their support, computational infrastructure, and human testing. This research was supported by the EU project "INPUT" (H2020-ICT-2015 grant no. 687795).

## Footnotes

[1]Note: This section only provides a short and high-level overview of the TAG framework and Tagger. For a more detailed description please refer to the supplementary material or the extended version of this paper: https://arxiv.org/abs/1606.06724

[2]The datasets and a Theano [33] reference implementation of Tagger are available at `http://github.com/CuriousAI/tagger`

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
