[Supplementary Material · tagger_supplementary.pdf]

# A  Supplementary Material

## A.1  Notation

| Symbol | Space | Description |
|---|---|---|
| $N$ | $\mathbb{N}$ | input dimensionality |
| $K$ | $\mathbb{N}$ | total number of groups |
| $H$ | $\mathbb{N}$ | input and output dimension of the parametric mapping |
| $i$ | $\mathbb{N}$ | iteration index |
| $j$ | $\{1,\ldots,N\}$ | input element index |
| $k$ | $\{1,\ldots,K\}$ | group index |
| $\boldsymbol{x}$ | $\mathbb{R}^N$ | input vector with elements $x_j$ |
| $\tilde{\boldsymbol{x}}$ | $\mathbb{R}^N$ | corrupted input |
| $\boldsymbol{z}_k$ | $\mathbb{R}^N$ | the predicted mean of input for group $k$ |
| $\boldsymbol{m}_k$ | $\mathbb{R}^N$ | probabilities for each input to be assigned to group $k$ |
| $\boldsymbol{\delta z}_k$ | $\mathbb{R}^N$ | modeling error for group $k$ |
| $L(\boldsymbol{m}_k)$ | $\mathbb{R}^N$ | group assignment likelihood ratio |
| $C(\boldsymbol{x})$ | $\mathbb{R}$ | the training loss for input $\boldsymbol{x}$ |
| $v$ | $\mathbb{R}$ | variance of the input estimate. Only used in the continuous case |
| $W_h$ | $\mathbb{R}^{H \times 4N}$ | Projection weights from tagger inputs to ladder inputs $\boldsymbol{h}$ |
| $W_u$ | $\mathbb{R}^{2N \times H}$ | Projection weights from ladder output to $\boldsymbol{z}$ and $\boldsymbol{m}$ |
| $\Theta$ | | Contains all parameters of the ladder |
| $f()$ | | rectified linear activation function |
| $g()$ | | logistic sigmoid activation function |
| $\mathrm{softmax}()$ | | elementwise softmax over the groups |
| $G_j$ | | Latent random variable that encodes which group $x_j$ belongs to. |
| $g_{k,j}$ | | Shorthand for $G_j = k$. Mostly used for $p(g_{k,j}) = p(G_j = k)$. |
| $\boldsymbol{g}$ | | a vector of all $g_j$. |
| $p(\boldsymbol{x} \mid \tilde{\boldsymbol{x}})$ | | posterior of the data given the corrupted data |
| $q(\boldsymbol{x})$ | | learnt approximation of $p(\boldsymbol{x} \mid \tilde{\boldsymbol{x}})$ |
| $q(x_j \mid g_{k,j})$ | | Shorthand for $q(x_j \mid G_j = k)$ |

## A.2  Input

In its basic form (without supervision) Tagger receives as input only a datapoint $\boldsymbol{x}$. It corresponds to either a binary vector or a real-valued vector and is then corrupted with either bitflip or Gaussian noise. The training objective is the removal of this noise.

**Bitflip Noise**   In the case of binary inputs we use bitflip noise for corruption:

$$\tilde{\boldsymbol{x}} = \boldsymbol{x} \oplus \mathcal{B}(\beta),$$

where $\oplus$ denotes componentwise XOR, and $\mathcal{B}(\beta)$ is Bernoulli distributed noise with probability $\beta$. In our experiments on the Shapes dataset we use $\beta = 0.2$.

**Gaussian Noise**   If the inputs are real-valued, we corrupt it using Gaussian noise:

$$\tilde{\boldsymbol{x}} = \boldsymbol{x} + \mathcal{N}(0, \sigma^2),$$

where $\sigma$ is the standard deviation of the input noise. We used $\sigma_{input} = 0.2$.

Figure 5: Illustration of the TAG framework used for training. Left: The system learns by denoising its input over iterations using several groups to distribute the representation. Each group, represented by several panels of the same color, maintains its own estimate of reconstructions $z^i$ of the input, and corresponding masks $m^i$, which encode the parts of the input that this group is responsible for representing. These estimates are updated over iterations by the same network, that is, each group and iteration share the weights of the network and only the inputs to the network differ. In the case of images, $z$ contains pixel-values. Right: In each iteration $z^{i-1}$ and $m^{i-1}$ from the previous iteration, are used to compute a likelihood term $L(m^{i-1})$ and modeling error $\delta z^{i-1}$. These four quantities are fed to the parametric mapping to produce $z^i$ and $m^i$ for the next iteration. During learning, all inputs to the network are derived from the corrupted input as shown here. The unsupervised task for the network is to learn to denoise, i.e. output an estimate $q(x)$ of the original clean input.

## A.3 Group Assignments

Within the TAG framework the group assignment is represented by the $K$ vectors $m_k$ which contain one entry for each input element or pixel. These entries $m_{k,j} = q(g_{k,j})$ of $m_k$ represent the discreet probability distribution over $K$ groups for each input $x_j$. They therefore sum up to one:

$$\sum_{k=1}^{K} m_{k,j} = 1 \quad \text{for all } j = 1 \ldots N \tag{2}$$

**Initialization**  Similar to expectation maximization, the group assignment is initialized randomly, but such that Equation 2 holds. So we first sample an auxiliary $m'_{k,j}$ from a standard Gaussian distribution and then normalize it using a softmax:

$$m'_{k,j} \sim \mathcal{N}(0,1) \tag{3}$$

$$m_{k,j} = \frac{e^{m'_{k,j}}}{\sum_{h=1}^{K} e^{m'_{h,j}}} \tag{4}$$

$$\tag{5}$$

## A.4 Predicted Inputs

Tagger maintains an input reconstruction $z_k$ for each group $k$.

**Binary Case**  In the binary case we use a sigmoid activation function on $z_k$ and interpret it directly as the probability

$$\text{sigmoid}(z_k) = q(x = 1 | g_k). \tag{6}$$

We can use it to compute $\tilde{z}_k = q(\tilde{x}|g_k)$ which will be used for the modeling error (Section A.5) and the group likelihood:

$$q(\tilde{x} = 1|g_k) = \sum_{x} q(\tilde{x}|x, g_k)q(x|g_k) \tag{7}$$

$$= \sum_{x} q(\tilde{x}|x)q(x|g_k) \tag{8}$$

$$= \sum_{x} \left(x(1-\beta) + (1-x)\beta\right)q(x|g_k) \tag{9}$$

$$= \sum_{x} \left(x(1-2\beta) + \beta\right)z_k \tag{10}$$

$$= \beta(1-z_k) + (1-\beta)z_k \tag{11}$$

$$= z_k(1-2\beta) + \beta \tag{12}$$

Therefore we have:

$$\tilde{z}_k = \tilde{x}(z_k(1-2\beta) + \beta) + (1-\tilde{x})(1 - z_k(1-2\beta) - \beta) \tag{13}$$

**Continuous Case**  For the continuous case we interpret $z_k$ as the means of an isotropic Gaussian with learned variance $v$:

$$q(x|g_k) = \mathcal{N}(x; z_k, vI) = \frac{1}{\sqrt{2\pi v}}e^{\frac{(x-z_k)^2}{2v}} \tag{14}$$

Using the additivity of Gaussian distributions we directly get:

$$\tilde{z}_k = q(\tilde{x}|g_k) = \mathcal{N}(\tilde{x}; z_k, (v + \sigma^2)I) \tag{15}$$

**Initialization**  For simplicity we initialize all $z_k$ to the expectation of the data for all $k$. In our experiments these values are $0.5$ for the TextureMNIST datasets and $0.26$ for the Shapes dataset.

## A.5 Modeling Error

As explained in Section 2, $\delta z$ carries information about the remaining modeling error. During training as a denoiser, we can only allow information about the corrupted $\tilde{x}$ as inputs but not about the original clean $x$. Therefore, we use the derivative of the cost on the corrupted input as helpful information for the parametric mapping. Since we work with the input elements individually we skip the index $j$ in the following:

$$\delta z_k \propto -\partial C(\tilde{x})/\partial z_k. \tag{16}$$

More precisely for a single iteration (omitting the index $i$) we have::

$$\delta z_k = -\frac{\partial C(\tilde{x})}{\partial z_k} \tag{17}$$

$$= \frac{\partial}{\partial z_k}\log\left(\sum_{h} q(\tilde{x}|g_h)q(g_h)\right) \tag{18}$$

$$= \frac{1}{\sum_{h} q(\tilde{x}|z_h)q(g_h)}\frac{\partial \sum_{h} q(\tilde{x}|z_h)q(g_h)}{\partial z_k} \tag{19}$$

$$= \frac{1}{\sum_{h} \tilde{z}_h m_h}\frac{\partial \tilde{z}_k}{\partial z_k}m_k \tag{20}$$

$$\tag{21}$$

**Continuous Case**  For the continuous case this gives us:

$$\delta z_k = \frac{1}{\sum_{h} \tilde{z}_h m_h}\frac{\partial \tilde{z}_k}{\partial z_k}m_k \tag{22}$$

$$= \frac{1}{\sum_{h} \tilde{z}_h m_h}\frac{\tilde{x} - z_k}{\sigma^2 + v}\tilde{z}_k m_k \tag{23}$$

$$\propto (\tilde{x} - z_k)m_k\tilde{z}_k \tag{24}$$

Note that since the network will multiply its inputs with weights, we can always omit any constant multipliers.

**Binary Case** Let us denote the corruption bit-flip probability by $\beta$ and define

$$\xi_k := q(\tilde{x} = 1 | g_k) = (1 - 2\beta)z_g + \beta \,.$$

Then we get:

$$\tilde{z}_k = \tilde{x}\xi_k + (1 - \tilde{x})(1 - \xi_k)$$

and thus:

$$\delta z_k = \frac{1}{\sum_h \tilde{z}_h m_h} \frac{\partial \tilde{z}_k}{\partial z_k} m_k \tag{25}$$

$$= \frac{(\tilde{x}(1 - 2\beta) - (1 - \tilde{x})(1 - 2\beta))m_k}{\sum_h (\tilde{x}\xi_h + (1 - \tilde{x})(1 - \xi_h))m_h} \tag{26}$$

which simplifies for $\tilde{x} = 1$ as

$$= \frac{(1 - 2\beta)m_k}{\sum_h \xi_h m_h} \approx -\frac{m_k}{\sum_h \xi_h m_h}$$

and for $\tilde{x} = 0$ as

$$= \frac{(1 - 2\beta)m_k}{1 - \sum_h \xi_h m_h} \approx \frac{m_k}{1 - \sum_h \xi_h m_h} = \frac{m_k}{\sum_h \xi_h m_h - 1}$$

Putting it back together:

$$\delta z_k = \frac{m_k}{\sum_h \xi_h m_h - 1 + \tilde{x}}$$

## A.6  Ladder Modifications

We mostly used the specifications of the Ladder network as described by Rasmus et al. [19], but there are some minor modifications we made to fit it to the TAG framework. We found that the model becomes more stable during iterations when we added a sigmoid function to the gating variable $v$ [19, Equation 2] used in all the decoder layers with continuous outputs. None of the noise sources or denoising costs were in use (i.e., $\lambda_l = 0$ for all $l$ in Eq. 3 of Ref. [19]), but Ladder's classification cost ($C_c$ in Ref. [19]) was added to the Tagger's cost for the semi-supervised tasks.

All four inputs ($z_k^i$, $m_k^i$, $\delta z_k^i$, and $L(m_k^i)$) were concatenated and projected to a hidden representation that served as the input layer of the Ladder Network. Subsequently, the values for the next iteration were simply read from the reconstruction ($\hat{x}$ in Ref. [19]) and projected linearly into $z_k^{i+1}$ and via softmax to $m_k^{i+1}$ to enforce the conditions in Equation 2. For the binary case, we used a logistic sigmoid activation for $z_k^{i+1}$.

## A.7 Pseudocode

In this section we put it all together and provide the pseudocode for running Tagger both on binary (Algorithm 3) and real-valued inputs (Algorithm 2). The provided code shows the steps needed to run for $T$ iterations on a single example $x$ using $G$ groups. Here we use three activation functions: $f(x) = \max(x,0)$ is the rectified linear function, $g(x) = \frac{1}{1+e^{-x}}$ is the logistic sigmoid, and $\text{softmax}(x)_g = \frac{e^{x_g}}{\sum_{h=1}^{G} e^{x_h}}$ is a softmax operation over the groups. All three include a batch-normalization operation, which we omitted for clarity. Only the forward pass for a single example is shown, but derivatives of the cost $C$ wrt. parameters $v$, $W_h$, $W_u$ and $\Theta$ are computed using regular backpropagation through time. For training we use ADAM with a batch-size of 100.

**Data**: $x, K, T, \sigma, v, W_h, W_u \Theta$
**Result**: $z^T, m^T, C$
**begin** Initialization:
$\quad \tilde{x} \leftarrow x + \mathcal{N}(\mathbf{0}, \sigma^2 \boldsymbol{I})$;
$\quad m^0 \leftarrow \text{softmax}(\mathcal{N}(\mathbf{0}, \boldsymbol{I}))$;
$\quad z^0 \leftarrow E[x]$;
**end**
**for** $i = 0 \dots T-1$ **do**
$\quad$ **for** $k = 1 \dots K$ **do**
$\quad\quad \tilde{z}_k \leftarrow \mathcal{N}(\tilde{x}; z_k^i, (v+\sigma^2)\boldsymbol{I})$;
$\quad\quad \boldsymbol{\delta z_k^i} \leftarrow (\tilde{x} - z_k^i) m_k^i \tilde{z}_k$;
$\quad\quad L(m_k^i) \leftarrow \frac{\tilde{z}_k}{\sum_h \tilde{z}_h}$ ;
$\quad\quad h_k \leftarrow f(W_h \left[ z_k^i, m_k^i, \boldsymbol{\delta z_k^i}, L(m_k^i) \right])$;
$\quad\quad [z_k^{i+1}, m_k^{i+1}] \leftarrow W_u \text{Ladder}(h_k, \Theta)$;
$\quad$ **end**
$\quad m^{i+1} \leftarrow \text{softmax}(m^{i+1})$;
$\quad q_{i+1}(x) \leftarrow \sum_{k=1}^K \mathcal{N}(x; z_k^{i+1}, v\boldsymbol{I}) m^{i+1}$;
**end**
$C \leftarrow -\sum_{i=1}^T \log q_i(x)$;

**Algorithm 2:** Pseudocode for running Tagger on a single real-valued example $x$.

**Data**: $x, K, T, \beta, W_h, W_u, \Theta$
**Result**: $z^T, m^T, C$
**begin** Initialization:
$\quad \tilde{x} \leftarrow x \oplus \mathcal{B}(\beta)$;
$\quad m^0 \leftarrow \text{softmax}(\mathcal{N}(\mathbf{0}, \boldsymbol{I}))$;
$\quad z^0 \leftarrow E[x]$;
**end**
**for** $i = 0 \dots T-1$ **do**
$\quad$ **for** $k = 1 \dots K$ **do**
$\quad\quad \boldsymbol{\xi_k} \leftarrow z^i(1 - 2\beta) + \beta$;
$\quad\quad \boldsymbol{\delta z_k^i} \leftarrow \frac{m_k^i}{\sum_h \xi_h m_h^i - 1 + \tilde{x}}$;
$\quad\quad L(m^i) \leftarrow \frac{\tilde{x}\xi_k + (1-\tilde{x})(1-\xi_k)}{\sum_h \tilde{x}\xi_h + (1-\tilde{x})(1-\xi_h)}$ ;
$\quad\quad h_k \leftarrow f(W_h \left[ z_k^i, m_k^i, \boldsymbol{\delta z_k^i}, L(m_k^i) \right])$;
$\quad\quad [z_k^{i+1}, m_k^{i+1}] \leftarrow W_u \text{Ladder}(h_k, \Theta)$;
$\quad$ **end**
$\quad m^{i+1} \leftarrow \text{softmax}(m^{i+1})$;
$\quad q_{i+1}(x) \leftarrow \sum_{k=1}^K \mathcal{N}(x; z_k^{i+1}, v\boldsymbol{I}) m^{i+1}$;
**end**
$C \leftarrow -\sum_{i=1}^T \log q_i(x)$;

**Algorithm 3:** Pseudocode for running Tagger on a single binary example $x$.