[Reviews · NeurIPS 2016]

Reviewer 1

Summary

Authors introduce a sequential denoising auto-encoder that splits inputs into groups (every pixel will approximately be assigned to a group). They demonstrate on artificial examples that the network is able to split inputs in the groups and that this achieves significant improvement in classification performance over common methods. Overall the work is interesting. However the drawback is that examples were specifically constructed as a perfect match for this network and it is not clear whether they would work well in real example, for example on a normal image. Also object separation is done in pixels. For generality we would like to see if this is useful in latent space and while authors mention that this can be done in latent, they don’t show any experiment.

Qualitative Assessment

The examples are rather artificial, they are non-trivial and the work is interesting. - It would be good to run the on real example - e.g. imageNet or some datasets which does contains occlusions - again showing separation and classification. The big question is whether this splitting is really helpful in real data. - It would be good to give an example of separation in latent space. Is this way of separating useful there? - It would be nice to write the full network with formulas in the paper - e.g. in equation array one after another (without the gradients - that is just computed using standard back-prop), including the classification part. It is hard to infer what exactly is the algorithm. For example you use ladder network, but how does it connect exactly to the m and z?

Confidence in this Review

2-Confident (read it all; understood it all reasonably well)


Reviewer 2

Summary

This paper extends the ladder network towards infering a layered hierarchical model of their inputs. More concretely, the network first infers a segmentation of the image in N groups (using pixel-wise probability masks over groups) which are then processed equally by a Ladder network. The inference- and generative steps are run multiple times in order to increase the denoising performance. The performance is evaluated on a shape and a texturized MNIST dataset.

Qualitative Assessment

UPDATE: I thank the authors for their convincing rebuttal, and in view of the promised updates on the technical specifications and description of the method, I increased the scores for "Technical quality" and "Clarity and presentation". My only major concern I still have is the lack of a suitable baseline to compare with. In particular, I do not agree that a comparison to [1] is impossible without their code. Instead, I'd encourage the authors to compare their method on the multi-MNIST benchmark described in Fig. 7 [1] (and to just use the numbers provided by [1] for comparison without re-simulation). This would significantly strengthen the results. --- The paper provides an interesting take on the old idea of layer models and the performance evaluation on the two datasets (shape and textures MNIST) indeed looks interesting. Unfortunately, however, I see two major flaws with the current presentation of the material: ** Literature and comparison to competitors First, the literature on this topic seems not to be suitably accounted for. E.g. a direct competitor that is based on Variational Autoencoders (VAE) [1] is noted as being similar in spirit but relying on MCMC sampling and not amortised inference. That is not true (amortized inference is at the heart of VAE approaches). Conversely, I believe that the performance of TAGGER should be directly compared to [1] e.g. on the overlapping MNIST task presented there. In addition, it is hinted that this work is related to the “grouping of oscillation” literature but the connection is left unclear. ** Style, Explanations, and Model Details Furthermore, there is little detailed and exact description of the TAGGER network, instead the reader is referred to the Ladder network literature and is then left to infer the additions and modifications from verbal instructions that are scattered throughout the paper and the supplementary material. Figure 2, which is meant to be the graphical description of the model, is not very helpful (e.g. what exactly is the “parametric mapping”? The ladder network encoding and decoding stage?). The verbal instructions are often confusing (where is the delta z in line 120 coming from?), not concise (e.g. no formula on decorrelation in SI) and lots of information on the exact learning paradigms are missing (e.g. learning rate). Some of this criticism is alleviated through the public code that is online (thumbs up to the authors!), but nonetheless the reader should not be required to go through the ladder network literature and the source code in order to understand what is going on. ** Minor - Add x-axis label in Fig. 3 - Please add a plot on the dependence of the classification performance in MNIST for different number of iterations [1] Efficient Inference in Occluion-Aware Generative Models of Images, Huang & Murphy

Confidence in this Review

2-Confident (read it all; understood it all reasonably well)


Reviewer 3

Summary

The paper proposes a framework of unsupervised learning based on iterative denoising. The framework is designed for a scenario where multiple labels are in the same input, such as image segmentation. The basic idea of the proposed approach is analogous to an EM-like approach to learn a mixture model, where grouping assignments (mixture coefficients) and components are inferred over alternating iterations. The proposed framework takes a corrupted input, and applies as amortized inference a denoising model to estimate the groups and their assignments. The obtained estimations are further fed into the next iteration. The learning objective is the negative log-likelihood of the approximation of the true posterior of denoting, which is modeled here by the mixture of groups. Using artificially generated Shapes and Textured MNIST datasets, the paper evaluates the clustering and semi-supervised classification performance. The empirical results indicate the proposed approach achieves significantly better result in both supervised and semi-supervised scenario, especially when multiple objects are present in the image.

Qualitative Assessment

The proposed approach looks unique and interesting. The framework combines the generative model with fast amortized inference to derive efficient iterative approach. The denoising objective naturally fits in the problem setting to make learning feasible both under semi-supervised and unsupervised scenario. This technical proposal looks novel and I believe highly contributing. The empirical results look also promising. Although the datasets are artificially generated and do not resemble a real-world situation as suggested in Fig 1, the significant performance improvement (Table 1) clearly indicates that the grouping is really helping in identifying multiple target objects in the same input. The experimental design is reasonable considering the purpose of the study. My biggest complaint is the lack of details in the paper. Sec 2 and 3 do not reveal enough detail of how the denoising objective and the parametric mapping is designed, and I had hard time understanding the model. Especially, m_BU in Fig 2 is never explained in the paper, and readers would have no clue what the mapping model takes as input and output. Also, although the objective is modeled after denoising, the proposed model is quite different from a standard denoising auto-encoder and the paper should not simply state ``we apply denoising auto encoders [20]’’ (Sec 2) since the model takes latent variables m and z from the previous iteration. I am curious why the paper states *perceptual* groups emerge out of the proposed model. The model seems to simply consider latent source signals in terms of denoising objective, and nothing constrains the groups. I can hardly imagine how the proposed model can lead to the segmentation example in Fig 1, or how the proposed model solves the ``dynamic, ambiguous and task-dependent’’ (Sec 1) grouping. To me, the proposed approach looks like a new, general generative framework based on denoising objective, and not tied to *perceptual* grouping as suggested in Sec 1. One related question is the robustness to initialization. The paper does not describe how the latent variables are initialized in the experiments nor how one should initialize m and z in general. Perhaps random initialization can work fine in the synthesized datasets, but I do not expect the same in much more complicated input like the segmentation example in Fig 1. Also, it is not clear if the proposed approach always converges. In overall, I would still give a positive rating. The paper is making a good contribution in proposing a novel clustering framework based on denoising formulation.

Confidence in this Review

2-Confident (read it all; understood it all reasonably well)


Reviewer 4

Summary

This paper presents a method for training a recurrent mixture density network using a denoising criterion. The intuition is that the mixture components will implicitly segment inputs into coherent perceptual/causal groups. Several experiments using synthetic data are presented in support of the proposed approach.

Qualitative Assessment

This paper develops a model which simultaneously segments and denoises an input using a recurrent mixture density network. The basic process feeds a noisy input into the network, after which several iterations of "amortized inference" are performed. The model outputs mixture means and weights, each of which has size D x K, where D is the input dimension and K is the number of mixture components. Additional inputs (basically, the gradient of denoising cost w.r.t. the current mixture means/weights) are fed into the network at each step to facilitate "inference". Figure 4 is a bit peculiar. The segmentations in the left columns appear sharp, yet the mixture weights in the detailed examples are messy. It would be helpful to show the step-by-step evolution of the combined mixture weights, to illustrate emergence of the crisp groupings. In the two-digit textured MNIST experiments, did you find that the assignment of top-left/bottom-right digit to a mixture component was stable across the test set? I.e., did a given trained model always assign the top-left digit to, e.g., component 1 and the bottom-right digit to component 2? For these tests, it would be more fair if two classifications were output by the baseline models, by using a pair of separate softmax layers which would be averaged prior to computing cross-entropy. The model description spanning lines 95-125 is a bit vague, and would benefit from either pseudo-code or a more precise mathematical description of the computations performed by the model. Sentences like the one spanning lines 120-122 are best avoided. They are confusing and irritating since, to anyone with basic competence in this area of research, the additional input described as "obvious" on line 121 is clearly non-sensical. Tests should be performed on one of the various multi-digit MNIST classification tasks that have been presented over the last 1-2 years (with or without clutter). I'm curious why the authors chose a denoising framework, rather than using a full Bayesian/probabilistic approach. E.g., one could run a non-attentive DRAW model or a recurrent VAE (fully-connected or convolutional), and just replace the output distribution with a mixture. The output format would be the same, but these methods would be more interpretable and would permit actual generation. A more fully probabilistic approach could also handle settings where the proper segmentation is ambiguous. Overall, this paper addresses an interesting problem domain and proposes a sensible model. It's hard to tell from the provided material whether the model is actually good at solving this problem. The presentation could be clearer and more succinct (making room for more thorough experiments), especially if the model was described as a direct extension of existing mechanisms.

Confidence in this Review

2-Confident (read it all; understood it all reasonably well)


Reviewer 5

Summary

Basing on TAG algorithm, the paper proposed a framework called iTAG to learn efficient iterative inference for perceptual grouping. It presented 4 improvement points and the algorithm is suitable for any kinds of data. It testd in two datasets and proved the improvements

Qualitative Assessment

Though the results seem to be inspiring, I think it is better to test its performance in more datasets

Confidence in this Review

1-Less confident (might not have understood significant parts)


Reviewer 6

Summary

The paper presents an unsupervised method of data grouping. The grouping is achieved by a mapping network, that tries to reconstruct the input given a noisy version of it, by combining many groups. The group assignment is iteratively refined. Experiments show that convergence of group assignment agrees with individual separation of objects.

Qualitative Assessment

In my personal opinion, the paper is difficult to read, at least for the nips submission version. The main idea is easy to catch, but after reading the first three sections three times, I still couldn't figure out the flow of the method. I happened to find the arXiv version of the paper, which has more detailed description and helped clear my confusions a lot. I suggest integrate those contents to the nips version. The method and its results are definitely interesting. I thus rate the paper poster level for novelty. The usefulness of the paper is difficult to assess. The experiments uses 20x20 shape images and 28x28 mnist images. How will the method perform in real-world scenarios remains a big question mark. To be specific, will a similar framework still be able to separate semantically different pixels in real-world images? Even if it somehow worked, how is it compared to simply stacking more layers? I am not convinced unless such experiments (even if in relatively small scale is fine) have been conducted.

Confidence in this Review

2-Confident (read it all; understood it all reasonably well)